# Mineral Concentrations in Different Types of Honey Originating from Three Regions of Continental Croatia

**DOI:** 10.3390/foods13172754

**Published:** 2024-08-29

**Authors:** Ivana Tlak Gajger, Damir Pavliček, Višnja Oreščanin, Ivana Varenina, Marija Sedak, Nina Bilandžić

**Affiliations:** 1Department for Biology and Pathology of Fish and Bees, Faculty of Veterinary Medicine, University of Zagreb, 10000 Zagreb, Croatia; itlak@vef.unizg.hr; 2Laboratory for Analytical Chemistry and Residues, Croatian Veterinary Institute, Veterinary Institute Križevci, Zakmardijeva 10, 48260 Križevci, Croatia; pavlicek.vzk@veinst.hr; 3Oreščanin Ltd., A. Jakšića 30, 10000 Zagreb, Croatia; vorescan@gmail.com; 4Laboratory for Residue Control, Department for Veterinary Public Health, Croatian Veterinary Institute, Savska Cesta 143, 10000 Zagreb, Croatia; kurtes@veinst.hr (I.V.); sedak@veinst.hr (M.S.)

**Keywords:** honey, botanical origin, environment factors, trace elements, EDXRF spectrometry

## Abstract

Honey has been recognized as a reliable indicator of environmental quality because of honeybees’ intense foraging activity, which brings them into contact with many persistent organic pollutants around the hive. In this study, four types of honey (meadow, acacia, chestnut, and honey in comb) collected at three different locations were analyzed for Co, Cr, Cu, Fe, Mn, Pb, and Zn levels. The highest levels of Fe and Cu in chestnut honey, Co and Zn in meadow honey, and Pb in honey in comb were observed in Varaždin County. The lowest levels of Pb in meadow honey and Co in comb honey were found from apiaries in Sisak-Moslavina County. Significant differences in the mean concentrations of Cr, Cu, Mn, and Fe were observed among the four honey types. Conversely, no significant differences in Co, Pb, and Zn levels were found. Most of the significant differences between the elements are related to chestnut honey. While sampling location (Fe) and type of honey (Pb), or both (Cr and Zn), significantly influenced the concentrations of some elements, these factors were found to be irrelevant for Mn, Co, and Cu. The results showed varying degrees of similarities and differences in mineral levels in honey samples, depending on floral and geographical origin.

## 1. Introduction

Honey is a unique natural product containing bioactive compounds derived from honeybees and plants [1]. The composition of honey is very complex and variable and depends on its botanical origin, as well as environmental and seasonal factors [2,3], and beekeeping practices (extraction, processing, and storage of honey) [4,5]. The composition and organoleptic characteristics of honey, such as color, taste, and texture, vary due to the content and origin of raw materials. These factors primarily depend on the floral origin of the nectar (monofloral or polyfloral) or honeydew collected by forager honeybees, in addition to the season and geographical location [6]. Minerals are minor constituents of honey (0.04 to 0.20% depending on the specific honey type) but play a crucial role in determining its quality. The mineral composition of honey is correlated with the color of the honey and its antioxidant properties. Additionally, the concentrations of minerals are associated with the electrical conductivity of honey [5].

Honey is known as an ancient remedy and is currently utilized in alternative medicine for treating various diseases and physiological disorders [7,8,9]. Several properties of honey, such as a high content of reducing sugars, high viscosity, low pH, high osmolarity, hydrogen peroxide production, and the presence of phytochemical components (phenolic acids, flavonoids, lysozyme, certain enzymes, ascorbic acid, proteins, and carotenoids) with antioxidant properties, contribute to its antimicrobial activity [10,11]. Additionally, due to its high nutritive and medicinal value, honey represents a quality food that can enhance the general health and immunity of consumers [12]. To achieve these positive health effects, honey must be free of toxic and harmful environmental contaminants or pollutants.

All beehive products and adult honeybees have been proposed as useful bioindicators of environmental health [13,14,15]. Mineral (macroelements, minor or trace elements, and heavy metals) content can be monitored by measuring their concentrations in honey [15,16,17]. Heavy metals mainly occur in the environment after their emission from a variety of anthropogenic sources or activities and consequently contaminate plant tissues, as well as pollen and nectar, which represent the main food for honey bees [5]. As honey bees typically forage within a range of 3 km from their apiaries, and in certain cases may extend their foraging distance to 8–10 km, they may collect a variety of elements (e.g., heavy metals) from contaminated nectar, pollen, and water while moving over flowers or dust particles during flight [6,13]. Due to the specific foraging biology, ecology, and behavior characteristics of honey bees, they bring collected contaminated food materials into the hive. In the hive, these materials are processed by house bees and stored in comb, where accumulated substances can be hazardous to the entire colony. After beekeepers harvest contaminated honeybee products, the metals enter the human food chain and can potentially cause adverse effects on consumers’ health [18]. On the other hand, some of the essential elements derived from plant and organic sources (micro or trace elements) are necessary compounds important for maintaining vital functions and the growth and development of organisms [19]. Due to the possible long persistence in the environment, cumulative nature, and negative effects of toxic metals, it is essential to constantly monitor the content of these pollutants in honey, which is a human food source [5].

According to European legislation, maximum residue levels (MRL) for elemental contaminants are established for food products, with a particular focus on contaminants like lead (Pb; 0.10 mg/kg) that are of significant concern to consumers of honey, either due to their toxicity or their potential presence in the ecosystem [20]. Therefore, information on the concentrations of elements in different types of honey and their potential impact on dietary intake is crucial for evaluating public health risks and assessing environmental health status.

In the present study, the concentrations of seven metal elements were analyzed in 36 samples of four types of honey collected from hives in apiaries located in three distinct geographical regions in continental Croatia. The analysis was conducted using the energy dispersive X-ray fluorescence (EDXRF) method. The objectives of this research were to determine and assess potential variations in the levels of cobalt (Co), chromium (Cr), copper (Cu), iron (Fe), manganese (Mn), Pb, and zinc (Zn) in different types of honey originating from plant pastures typical for the continental part of the country (meadow, acacia, chestnut, and mixed honey in comb) and located near urban industrial zones.

## 2. Materials and Methods

### 2.1. Sampling

A total of 36 honey samples, including meadow, acacia, chestnut, and mixed honey in comb (12 samples per location, 3 per type), were collected from stationary apiaries in the continental region of Croatia, specifically in the counties of Varaždin, Zagreb, and Sisak-Moslavina (Figure 1). The apiaries were located in diverse geographical and environmental settings. Location 1 was situated in grassland surrounded by fields used for intensive agriculture, near a highway and a village. Location 2 was in a rural area surrounded by orchards and vineyards. Location 3 was in an urban area, on the border of an industrial zone with intensive refinery activity.

For each apiary, samples were taken from three randomly chosen standard Langstroth hives (LR hives). During the active beekeeping season (May–July; September) four types of honey were sampled from the peripheral frames of the brood hive chamber: meadow (M, a mix of flowers); acacia (A, *Robinia pseudoacacia*); chestnut (C, *Castanea sativa*); and honey in comb (HC, mixed honey origin). Samples of honey in comb are, in fact, honey that has not been extracted; instead, they are harvested along with a portion of the honeycomb.

Upon collection, all honeybee samples were placed into clean glass containers, labeled, and transferred to the laboratory. They were kept at 4 to 8 °C until analysis.

### 2.2. Sample Preparations, Chemicals, Standards, and Analyses

To obtain homogenized honey samples, containers with approximately 200 g of each sample were carefully melted in a water bath at approximately 40 °C and sonicated [21]. Subsequently, 1 g of each honey sample was transferred into Teflon vessels, and 5 mL of an HNO_3_-H_2_O_2_ mixture (4:1) was added for digestion in a microwave oven Multiwave 3000 (Anton Paar, Graz, Austria). After cooling to room temperature, the samples were diluted with deionized water to a total volume of 100 mL. The solution was then adjusted to pH 3 with 36% (*w*/*w*) hydrochloric acid for the analysis of Cu, Fe, Pb, and Zn and to pH 12 with ammonium hydroxide for the analysis of Co, Cr, and Mn. pH measurements were conducted using a digital pH meter (Mettler Toledo, Columbus, OH, USA). Following pH adjustments, 2 mL of 1% (*w*/*v*) ammonium-pyrroloidinedithiocarbamate (APDC) (Merck, Schuchardt, Germany) was added to each flask for complexation, which lasted for 20 min. The suspension was then filtered through a Millipore MF-Millipore™ mixed cellulose ester membrane filter with a pore size of 0.45 μm and a diameter of 25 mm using the Millipore microfiltration system (Millipore, Billerica, MA, USA). The prepared thin targets were air-dried, protected by a thin mylar foil (2 μm), inserted into a plastic carrier, and loaded into the X-ray spectrometer.

To measure the concentrations of metals in all honey samples, the source-excited energy dispersive X-ray fluorescence method [22] was employed. The prepared samples were irradiated with X-rays generated from an Rh tube (power: 9 W; window: 75 μm Be; high voltage: 30 kV; current: 300 μA; cooling medium: air). To reduce background interference, a 100 μm thick Ag filter was placed between the source and the prepared sample. Each sample was measured for 300 s, and the measurements were conducted in an air medium. The characteristic X-ray radiation from each sample was detected using a Si drift detector (surface: 5 mm^2^; FWHM for 5.9 keV 55Fe: 145 eV; window: 13 μm Be; cooling: thermo-electrical (Peltier)). The incident and emerging angles were set at 45°. The spectral data were analyzed using MiniPal/MiniMate software version 3.0–63 (2.64) (PANalytical, Almelo, The Netherlands). A spinner system was utilized to ensure constant rotation of the samples during measurement to minimize errors arising from non-homogeneity in sample preparation. The instrumental settings’ performance was automatically recorded every hour. An Al-Cu alloy served as the reference material for this purpose. AlKα and CuKα X-ray lines from the plate were employed for energy scale corrections, and a CuKα X-ray line was used for automatic gain control.

For qualitative and quantitative analyses, a calibration model was created based on measurements of standard solutions (Merck) in a concentration range from 10 to 200 μg/L. The preparation and measurements of all samples were conducted in the same manner as for the unknown samples. Blank measurements were run with deionized water from a Milli-Q Plus system (>18.3 MΩ) (Millipore, Billerica, MA, USA). To correct the contribution of the Zn results, deionized water was measured ten times, and the mass fractions obtained were recorded.

To verify the accuracy and reliability of the method, a quality control sample was prepared. A solution containing 100 μg/L of each target element was pre-concentrated with 2 mL APDC, filtered, and analyzed as a blind sample. The average recovery values (μg/L) obtained were as follows: Co (96 ± 1), Cr (98 ± 1), Cu (99 ± 2), Fe (99 ± 2), Mn (98 ± 2), Pb (97 ± 2), and Zn (99 ± 1). The limit of detection (LOD) was set at 1 μg/L for all observed metals. LOD values were established at one-third of the lowest mass fraction of the concentration for which the criteria for trueness (60–115%) and accuracy (RSD ≤ 25%) were satisfied. For the sake of simplicity, in terms of method performance and monitoring, the limits are set to the same value.

### 2.3. Statistical Analyses

Statistical evaluation was conducted using the STATISTICA 11.0 software package. Basic statistical parameters were calculated using standard statistical methods. The difference in the elemental concentrations among different types of honey and sampling locations was assessed using the Newman–Keuls test. The impact of predictor variables (types of honey, sampling location) on the concentrations of selected metals was examined using multiple regression analysis and the general regression model. The significance level for all tests was set at *p* < 0.05.

## 3. Results and Discussion

Although there are minerals naturally occurring in the biosphere that are essential for regulating metabolic pathways in living organisms, micro- or trace minerals originating from inorganic or metallic sources can contain potentially toxic substances [23]. These heavy metals, when present at low concentrations, tend to accumulate in organisms, contaminating food chains and potentially causing severe detrimental effects on human health. Honey is a widely consumed natural food, especially among children, and the presence of xenobiotic pollutants in honey can reduce its quality and health benefits [24].

The results of multi-element analysis of measured concentrations of Cr, Mn, Fe, Co, Cu, Zn, and Pb in samples of different honey types obtained from honeybee hives in Varaždin, Zagreb, and Sisak-Moslavina counties are presented in Table 1. Mean values of certain elements in relation to the type of honey and sampling location ranged from 0.22 to 10.1 mg/kg. Results indicate that the most abundant element was Fe, while Pb had the lowest mean values across all types of honey samples and locations. In fact, elements were found in the following descending order: Fe, Zn, Cu, Mn, Cr, Co, and Pb. The highest mean concentrations of Fe (10.1 mg/kg), Co (0.78 mg/kg), Cu (3.01 mg/kg), Zn (2.68 mg/kg), and Pb (0.40 mg/kg) were observed in samples taken from Location 1 (Varaždin), Mn (2.01 mg/kg) in samples taken from Location 2 (Zagreb), and Cr (1.49 mg/kg) in samples taken from Location 3 (Sisak-Moslavina). Additionally, when comparing the honey types, these maximum values were evenly distributed between meadow (Cr, Co, and Zn) and chestnut (Mn, Fe, and Cu), while Pb was detected at the highest level in honey from the comb. The lowest mean concentrations of Fe (4.57 mg/kg), Co (0.42 mg/kg), Cu (0.77 mg/kg), Zn (1.43 mg/kg), and Pb (0.22 mg/kg) were observed in samples from apiaries at Location 3 (Sisak-Moslavina). The minimum mean concentrations of Cr and Mn were observed in Varaždin samples.

Statistical analysis revealed significant differences in the mean concentrations of honey types, specifically for Cr, Cu, Mn, and Fe (Table 1). For Co, Pb, and Zn, no significant differences were found. Honey in comb located in Locations 2 and 3 exhibited significantly higher levels of Cr compared to honey from Location 1. Additionally, at Location 1, a significantly higher concentration of Cr was found in meadow honey (1.46 mg/kg) compared to honey in comb (0.65 mg/kg). Statistically significant differences in Mn concentrations were observed at Location 1, where chestnut honey (1.13 mg/kg) had a significantly higher concentration than acacia honey (0.69 mg/kg). Conversely, at Location 2, acacia honey exhibited higher concentrations (0.95 mg/kg) compared to chestnut honey (0.80 mg/kg). Furthermore, significantly higher concentrations of Mn were determined in chestnut honey at Location 2 (2.01 mg/kg) compared to those from Locations 1 and 3.

Significantly elevated concentrations of Cu and Fe were found in chestnut honey at Location 1 (3.01 mg/kg and 10.1 mg/kg, respectively) compared to Locations 2 (0.87 mg/kg and 5.15 mg/kg) and 3 (0.98 mg/kg and 5.65 mg/kg). The concentrations of Cu and Fe in chestnut honey from Location 1 were significantly higher than those in meadow, acacia, and honey in comb of the same location. Acacia and chestnut honey from Location 3 had significantly higher content of Fe compared with honey in comb. Also, the content of Fe in honey in comb from Location 1 was significantly higher compared with samples from Location 3.

Results of multiple regression analysis showed that the predictor variables had good, statistically significant influence on the concentrations of the elements Cr (R = 0.51; *p* < 0.007), Fe (R = 0.42; *p* < 0.041), Zn (R = 0.61; *p* < 0.0004), and Pb (R= 0.65; *p* < 0.0001). Based on the beta coefficients and their significance, both the location and the type of honey contributed significantly to the concentrations of Cr and Zn. In the case of Fe, only the sampling location had a significant influence. However, in the case of Pb, only the type of honey contributed significantly to its concentrations.

The results of the general regression model, expressed as Pareto charts of t-values (Figure 2), were completely in agreement with those obtained using multiple regression analysis. Both the location and the type of honey had a statistically significant influence on the concentrations of Cr and Zn. The location had a somewhat higher influence, which was also consistent with the beta coefficients obtained from multiple regression analysis. The location significantly influenced the concentrations of Fe, while the type of honey influenced the concentration of Pb. Neither the sampling location nor the type of honey made a statistically significant contribution to the concentrations of Mn, Co, and Cu.

The concentrations of most elements in different types of honey from various locations were relatively evenly distributed. However, exceptions were observed in chestnut honey with higher proportions of Fe and Cu at Location 1, meadow honey with higher Zn levels at Location 1, and chestnut honey with elevated Mn levels at Location 2 compared to other data. The increased concentrations of these elements found in chestnut honey were anticipated in this dark variety, which is known to be richer in minerals [25]. As is well known, the composition of elements in honey varies based on its botanical and geographical origins [5,17].

Relatively consistent mean concentrations were found for Cr and Mn. The lowest concentrations for both trace elements were detected in honey samples from Varaždin county (Location 1), Cr in honey in comb, and Mn in acacia honey. Conversely, the highest Cr content was identified in meadow honey from Sisak-Moslavina county (Location 3) and Mn content in chestnut honey from Zagreb (Location 2). When compared to findings from other studies, where the analyzed honeys were produced in densely populated metropolitan areas or regions with agricultural and forestry activities, the mean Cr levels were generally higher, while Mn levels were lower than those reported in the literature [26,27]. This observation is further supported by comparing the results with a previous report on Cr and Mn content in Croatian honey of various botanical origins [28,29].

Iron was the most prevalent element in this study, exhibiting similar concentrations across all honey types, except for the highest value found in chestnut honey from Varaždin County (Location 1). Detected Fe values were similar to or moderately higher than those in honey samples from other studies conducted in continental Croatia [28,30]. In chestnut honeys from Spain, lower levels of Fe were found at 7.07 mg/kg [31]. In contrast, a higher Fe content of 24.0 mg/kg was determined in multifloral honey from Poland [32]. In a recent study from Montenegro, Fe concentrations were measured in the range of 3.95–15.93 mg/kg [33]. In acacia honey from Romania, higher values than those found in this study, ranging from 15.98 to 32.05 [34], were observed.

The mean copper concentrations ranged from 0.77 mg/kg in acacia honey from Location 3 (Sisak-Moslavina) to 3.01 mg/kg in chestnut honey from Varaždin county. These measured levels of Cu were slightly higher than those found in honeys from different geographical regions of Australia (0.05–4.8 mg/kg, mean value 0.2 mg/kg) [26], Pakistan (0.08–0.33 mg/kg) [35], and Poland (0.05–1.38 mg/kg) [36], but lower than levels previously reported in Croatia (4.38–20.6 mg/kg) [16] and Slovenia (0.37–15.5 mg/kg) [37].

Comparing the Co content in honey samples, it can be concluded that neither the sampling site nor the type of honey has a significant effect on the final proportion of the element. Although average Co concentrations were among the lowest in this study, the levels obtained were still higher than those reported in some other studies from Croatia. Bilandžić et al. [29] measured Co above LOD values in only 8.6% of acacia honey, with levels ranging between 23 and 95 μg/kg, and in 19.6% of chestnut honey, with levels ranging between 7.3 and 17 μg/kg. Similar results were obtained by Lanjwani and Channa [35], who reported a mean Co concentration of 0.06 mg/kg (min–max: 0.01–0.23 mg/kg). However, in a study by Tutun et al. [38], Co was not detected in 70 honey samples obtained from beekeepers located in the West Mediterranean region of Turkey.

In the present study, the second most abundant mineral in honey samples was Zn. The Zn content was consistent across all analyzed honey samples from three different locations. Bilandžić et al. [29] investigated the concentrations of elements in floral, acacia, and chestnut honey collected from various regions of Croatia during the 2013, 2015, and 2016 seasons. Zinc values were reported in the range of 0.86 to 3.8 mg/kg (floral), 0.30 to 1.7 mg/kg (acacia), and 0.76 to 1.4 mg/kg (chestnut). Similar levels of Zn have been detected in other countries, such as Chile [39], Kosovo [27], Pakistan [35], Poland [36], and Slovakia [40]. In a study by Hungerford et al. [26], relatively higher levels of Zn ranging from 0.16 to 120 mg/kg were found, with rural honey showing higher values compared to urban honey.

Previous studies have suggested a significant influence of soil composition on the microelement content in honeys [41]. However, the geochemical composition of soil in the studied locations in continental Croatia showed no significant differences in content of Co, Cr, Cu, Fe, Mn, and Zn [42]. Statistical analysis indicated that only the levels of Cr and Zn in honey were significantly influenced by the location and type of honey. However, in this study, botanical origin has the most significant influence on element concentrations. The influence of geographical origin was only observed in the concentrations of Cu, Fe, and Mn in chestnut honey. External factors such as environmental and agricultural sources of contamination can significantly influence the results.

The results of the Pb content analysis in honey samples revealed concerning levels of this toxic metal. Mean concentrations varied from 0.22 mg/kg in meadow honey from Sisak-Moslavina County to a maximum value of 0.40 mg/kg in comb honey in Varaždin. Lead content exceeded the set MRL value of 0.1 mg/kg [20] in all analyzed samples. These values were also observed in honey sourced from rural areas where hives are situated at a considerable distance from highways, indicating that there is no evidence of pollution from internal combustion engine vehicles. One possible explanation is the high concentration of Pb in the soil of Varaždin County, where the highest Pb concentrations in Croatia were measured (up to 699 mg/kg), several times higher than the average value (83 mg/kg) for the area of Croatia [42]. In the study by Bilandžić et al. [43], the mean value of Pb content in honey samples from five different regions of Croatia was 65 µg/kg. The lowest measured value was 10 µg/kg, while the highest concentration of 841 µg/kg was obtained in multifloral honey from Northeast Croatia. A similar study of element contents in floral, acacia, and chestnut honey in Croatia showed somewhat lower mean concentrations of Pb, ranging from 5.1 to 20 µg/kg, <2 to 14 µg/kg, and 6.7 to 33 µg/kg, respectively [29]. The levels observed in this study were comparable to those found in Kosovo (0.235–0.268 mg/kg) [44], Greece (0.26–0.41 mg/kg) [45], and Iraq honey (0.100–0.730 mg/kg) [46], whereas the Pb content in analyzed honey samples from Montenegro (0.01–0.21 mg/kg) [33] and Italy (0.001–0.064 mg/kg) [47] is significantly lower. Furthermore, in honey from Iran (0.117–1.63 mg/kg) [48] and Romania (0.76–3.41 mg/kg) [34], concentrations of Pb were found to be several times higher than the maximum recommended by EU legislation [20].

## 4. Conclusions

This study provides insights into the distribution of trace elements in honey collected from continental Croatia. In most cases, the concentrations of elements in the various types of honey from the three locations were relatively evenly distributed. Iron exhibited the highest concentration, while Pb had the lowest levels. Maximal values were distributed between meadow honey (Cr, Co, and Zn) and chestnut honey (Mn, Fe, and Cu), while the highest Pb values were detected in honey in comb. Furthermore, significant differences in the mean concentrations of Cr, Cu, Mn, and Fe among four honey types were observed. Conversely, no significant differences were found for Co, Pb, and Zn. Most of the significant differences between the elements are related to chestnut honey. The highest levels of Fe and Cu were found in chestnut honey. Co and Zn levels were elevated in meadow honey, and Pb was most concentrated in honey in comb from Varaždin County (Location 1). The elevated levels of trace elements in honey may have been influenced by anthropogenic activities, including environmental and agricultural sources of contamination, as well as the soil composition in microlocations.

Although apiaries near highways and urban areas are more likely to accumulate higher levels of Pb compared to those in rural areas, this study found relatively similar levels of Pb in samples from all three sites. However, all measured Pb concentrations exceeded the MRL establish by the EU, raising concerns about the toxicological risks to honeybee colonies. Given the toxic nature of Pb, the obtained concentrations highlight the importance of continuous monitoring of its levels.

## Figures and Tables

**Figure 1 foods-13-02754-f001:**
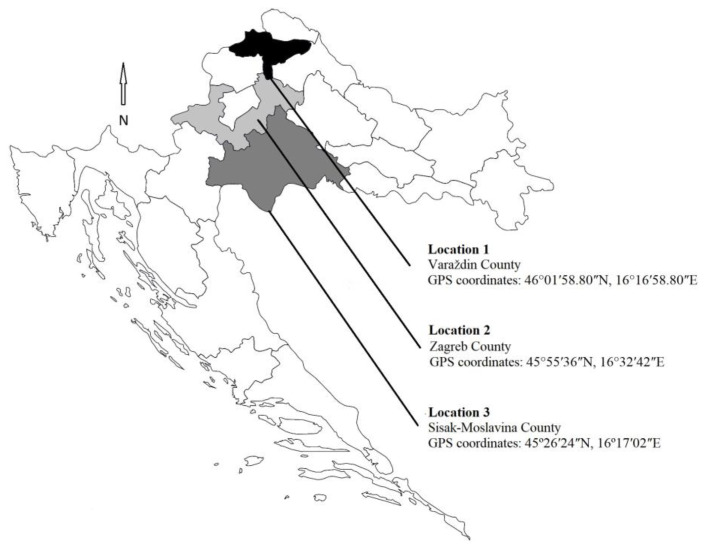
Location sampling sites where a total of 36 pooled samples of different honey types were collected in the continental part of Croatia (Location 1, Location 2, and Location 3) in the counties of Varaždin, Zagreb, and Sisak-Moslavina.

**Figure 2 foods-13-02754-f002:**
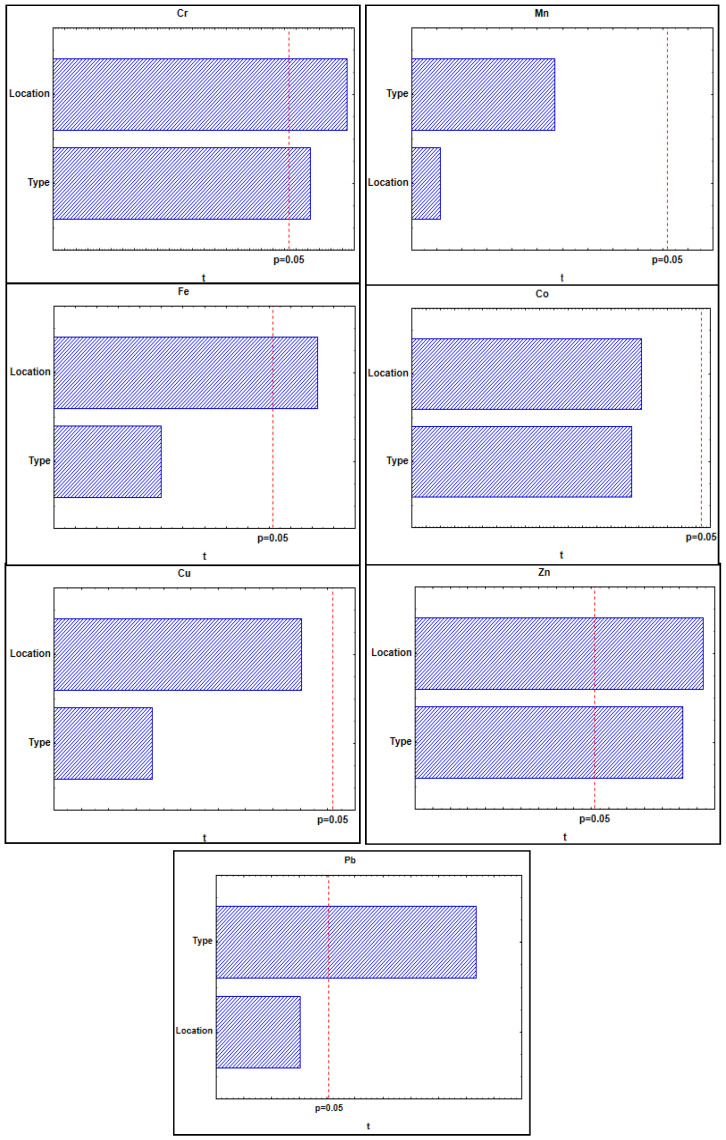
Results of general regression model expressed as pareto charts of t-values obtained for the influence of predictor variables on the concentrations of selected elements determined in honey samples. Marked differences are significant at *p* < 0.05.

**Table 1 foods-13-02754-t001:** Concentrations of metals (mg/kg) measured in samples of various types of honey from three apiaries using the EDXRF method.

Element	Statistic	Location 1/Type of Honey				Location 2/Type of Honey				Location 3/Type of Honey			
		M	A	C	HC	M	A	C	HC	M	A	C	HC
Co	Mean	0.78	0.62	0.65	0.69	0.66	0.70	0.75	0.67	0.72	0.55	0.74	0.46
	SD	0.08	0.09	0.06	0.05	0.06	0.07	0.05	0.21	0.04	0.08	0.06	0.04
	Min	0.70	0.56	0.60	0.65	0.60	0.65	0.70	0.45	0.69	0.50	0.70	0.42
	Max	0.86	0.72	0.72	0.75	0.71	0.78	0.80	0.86	0.76	0.64	0.80	0.50
Cr	Mean	1.46	0.97	1.19	0.65 ^b^	1.03	1.26	0.79	1.27 ^b^	1.49	1.28	1.31	1.27 ^b^
	SD	0.29	0.29	0.03	0.06	0.29	0.05	0.03	0.28	0.16	0.03	0.08	0.21
	Min	1.28	0.65	1.16	0.59	0.80	1.20	0.76	0.98	1.40	1.24	1.24	1.10
	Max	1.80	1.20	1.22	0.70	1.35	1.30	0.82	1.54	1.67	1.30	1.40	1.50
Cu	Mean	1.05 ^a^	0.84 ^a^	3.01 ^ab^	0.94 ^a^	1.01	1.05	0.87 ^b^	1.00	1.12	0.77	0.98 ^b^	1.17
	SD	0.19	0.15	0.08	0.12	0.20	0.83	0.09	0.30	0.29	0.02	0.02	0.21
	Min	0.90	0.71	2.92	0.80	0.87	0.10	0.80	0.70	0.90	0.76	0.97	1.00
	Max	1.26	1.00	3.08	1.02	1.24	1.60	0.98	1.30	1.45	0.80	1.00	1.40
Fe	Mean	5.13 ^a^	5.22 ^a^	10.1 ^ab^	5.87 ^ab^	5.24	5.58	5.15 ^b^	5.63	5.24 ^a^	5.61 ^a^	5.65 ^ab^	4.57 ^ab^
	SD	0.14	0.09	0.14	0.81	0.14	0.29	0.07	0.54	0.32	0.27	0.28	0.45
	Min	5.00	5.13	9.98	5.24	5.12	5.34	5.10	5.00	5.00	5.30	5.34	4.10
	Max	5.28	5.30	10.2	6.78	5.40	5.90	5.23	5.98	5.60	5.80	5.90	5.00
Mn	Mean	0.88	0.69 ^ab^	1.13 ^a^	0.82	0.93	0.72	2.01 ^b^	0.79	0.80	0.95	0.80 ^b^	1.10
	SD	0.09	0.17	0.08	0.02	0.27	0.09	0.10	0.02	0.05	0.14	0.04	0.26
	Min	0.80	0.53	1.04	0.80	0.70	0.64	1.90	0.77	0.76	0.84	0.76	0.90
	Max	0.97	0.86	1.20	0.84	1.23	0.82	2.10	0.80	0.85	1.10	0.84	1.40
Pb	Mean	0.26	0.30	0.38	0.40	0.24	0.31	0.37	0.35	0.22	0.28	0.39	0.30
	SD	0.05	0.09	0.08	0.02	0.05	0.11	0.05	0.05	0.02	0.02	0.03	0.05
	Min	0.20	0.22	0.31	0.39	0.20	0.20	0.32	0.30	0.20	0.26	0.36	0.25
	Max	0.30	0.40	0.46	0.42	0.30	0.42	0.41	0.40	0.24	0.30	0.41	0.35
Zn	Mean	2.68	1.56	1.99	1.69	1.60	1.67	1.57	1.63	1.92	1.43	1.65	1.43
	SD	0.06	0.18	0.10	0.04	0.09	0.03	0.03	0.10	0.06	0.08	0.18	0.21
	Min	2.64	1.42	1.90	1.65	1.53	1.65	1.54	1.54	1.87	1.36	1.50	1.20
	Max	2.75	1.76	2.10	1.72	1.70	1.71	1.60	1.74	1.98	1.52	1.85	1.60

SD—standard deviation; M—meadow; A—acacia; C—chestnut; HC—honey in comb; ^a^ Significant differences between honeys within same location; ^b^ Significant differences in same type of honeys between locations.

## Data Availability

The original contributions presented in the study are included in the article, further inquiries can be directed to the corresponding author.

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
