# Peer review of "Mineral Concentrations in Different Types of Honey Originating from Three Regions of Continental Croatia"

_foods, 2024, doi:10.3390/foods13172754_

Round 1

Reviewer 1 Report

Comments and Suggestions for Authors

“Mineral concentrations in different types of honey originating from locations along an urban-rural gradient” is an interesting study, especially since the authors have used instrumental method which is not so often applied to the food samples.

Introduction is very informative and MM section is sound and precisely written. I really like the idea behind this research but I have some serious issues with design of the study, methods and interpretation of the results.

The authors made an effort to controlled accuracy and reliability with quality control program which included recovery tests. They did not report that they were using blanks and I also find problematic statement within lines 157-158 about detection limit. Each element should have different detection limit, so how was it possible to set it at exactly 1 μg/L for all of them?

Another major issue is that no certified reference materials were used. The authors have reported some unexpected results for several elements such as Co and Pb which are order of magnitude higher than in other studies. I can not be sure for what is the cause, maybe these elements were really present in the samples, but maybe they are elevated due to instrumental bias. I am not sure what could be exact cause for cobalt elevation (interference with other elements since it is next to Fe in PSE or maybe it is present in the equipment), but for Pb it is well known that it has overlapping line with S which is also present in used extraction agent. The results for Pb also have huge standard deviation (5 to 30%) which means that they are unreliable due to interferences or due to proximity to detection limit.

The next problem with the design of the study is the small number of samples within one group of samples (those which are taken at the same place and with the same type of honey) which is limited to only 3 per group. This makes statistical analysis very challenging, and I guess that is why authors used less conservative Newman-Keuls test. Nevertheless it seems that we would get similar conclusions even if we double the number of the samples within a group, since there seems to be no trend either if we analyse types of honey or different locations.

The problem with honey analysis is related to the fact that bees try to minimalize adverse effects of the environment, therefore the choice of locations is not the best since it e.g. does not include no industrial sources focused on the metal processing. I don’t think that use of term “urban-rural gradient” is justifiable since no gradient is seen in results, but also can not be seen in the design of the study.

In my opinion, the results which are explaining that no major differences are noticed even in situations where we would expect to see one are equally scientifically relevant and I would suggest that discussion should be changed more in that direction. It is all right if you did not find trends you were expecting or hoped to find.

 I would also suggest to further investigate if some differences (e.g. for Co or Pb) could be the consequence of different instrumental techniques (e.g. ICP-MS or ICP-OES vs XRF). Implication that XRF has potential drawbacks compared to the alternatives could have more value than attempts to analyse data (Pb in honey) burdened with high load of noise.

Minor remarks:

l111-112 can you please clarify what is the difference between “honey in combs (HC, mixed honey origin)” and other types of honey? What does mixed honey origin mean?

l191 quality of Fig1. is low, and I would recommend to put all locations next to each other (in the same horizontal line)

Comments on the Quality of English Language

l85 please add “lead” in front of Pb

l127 Pyrrolidine is missing one r ?

Author Response

Mineral concentrations in different types of honey originating from locations along an urban-rural gradient” is an interesting study, especially since the authors have used instrumental method which is not so often applied to the food samples.

Introduction is very informative and MM section is sound and precisely written. I really like the idea behind this research but I have some serious issues with design of the study, methods and interpretation of the results.

The authors made an effort to controlled accuracy and reliability with quality control program which included recovery tests. They did not report that they were using blanks and I also find problematic statement within lines 157-158 about detection limit. Each element should have different detection limit, so how was it possible to set it at exactly 1 μg/L for all of them?

Explanation were added in text:

L155-158: Blank measurements were run with deionised water from a Milli-Q Plus system (> 18.3 MΩ) (Millipore, Billerica, MA,USA). To correct the contribution of the Zn results, deionised water was measured ten times and the mass fractions obtained were recorded.

L164-167: Sentences is added in text: LOD values were established at one-third of the lowest mass fraction of the concentration for which the criteria for trueness (60 - 115%) and accuracy (RSD ≤ 25%) were satisfied. For the sake of simplicity, in terms of method performance and monitoring, the limits are set to the same value.

Another major issue is that no certified reference materials were used. The authors have reported some unexpected results for several elements such as Co and Pb which are order of magnitude higher than in other studies. I can not be sure for what is the cause, maybe these elements were really present in the samples, but maybe they are elevated due to instrumental bias. I am not sure what could be exact cause for cobalt elevation (interference with other elements since it is next to Fe in PSE or maybe it is present in the equipment), but for Pb it is well known that it has overlapping line with S which is also present in used extraction agent. The results for Pb also have huge standard deviation (5 to 30%) which means that they are unreliable due to interferences or due to proximity to detection limit.

Explanation: Since there were no certified reference materials of honey available, quality control sample was prepared as described in the article. The mean values for all elements were within acceptable recovery range (60 - 115 %). Concentration of Co were higher than in some other studies in Croatia, but this number is quite limited in terms of this element, and if we take into account the number of samples taken from three specific locations, the values ​​obtained seem plausible. Regarding the results for Pb, standard deviation is in the range 2-11 % not 5-30% like stated. In our case, higher STDEV values would not be unexpected since the standard deviation of the distribution of sample means decreases with the increase of the sample size (number). Also, Pb concentrations obtained in the samples were far above LOD, and possible explanation of these results and comparison with other similar domestic and international studies was fairly described in the paper.

The next problem with the design of the study is the small number of samples within one group of samples (those which are taken at the same place and with the same type of honey) which is limited to only 3 per group. This makes statistical analysis very challenging, and I guess that is why authors used less conservative Newman-Keuls test. Nevertheless it seems that we would get similar conclusions even if we double the number of the samples within a group, since there seems to be no trend either if we analyse types of honey or different locations.

Explanation: Regarding the number of samples, the reviewer is right, that's why the less conservative Newman-Keuls test was used.

The problem with honey analysis is related to the fact that bees try to minimalize adverse effects of the environment, therefore the choice of locations is not the best since it e.g. does not include no industrial sources focused on the metal processing. I don’t think that use of term “urban-rural gradient” is justifiable since no gradient is seen in results, but also can not be seen in the design of the study.

Answer: we agree with the reviewer's comment and accordingly we will change the title of the paper, which is actually more precise regarding geographical locations: Mineral concentrations in different types of honey originating from three regions of continental Croatia

Line 98-99: sentence is modified to: The apiaries were located in diverse geographical and environmental settings.

In my opinion, the results which are explaining that no major differences are noticed even in situations where we would expect to see one are equally scientifically relevant and I would suggest that discussion should be changed more in that direction. It is all right if you did not find trends you were expecting or hoped to find.

L203-220: Results of statistics were in details showed in table 1. Tables 2-4 are excluded from article. The statistical results have been verified. The results for Cu, which were omitted by mistake, have been inserted. Statistics showed also comparison of different honey types per location. In Table 1 are added a- Significant differences between honeys within location; b- Significant differences in same type of honeys between locations. Statiastical results were added in text: „Statistical analysis revealed ....“

L304-313: Answer for explanation for obtained results sentences are added (sentences is in purple letters): Previous studies have suggested a significant influence of soil composition on the microelement content in honeys [41]. However, the geochemical composition of soil in the studied locations in continental Croatia showed no significant differences in content of Co, Cr, Cu, Fe, Mn and Zn [42]. Statistical analysis indicated that only the levels of Cr and Zn in honey were significantly influenced by the location and type of honey. However, in this study, botanical origin has the most significant influence on element concentrations. The influence of geographical origin was only observed in the concentrations of Cu, Fe, and Mn in chestnut honey.

I would also suggest to further investigate if some differences (e.g. for Co or Pb) could be the consequence of different instrumental techniques (e.g. ICP-MS or ICP-OES vs XRF). Implication that XRF has potential drawbacks compared to the alternatives could have more value than attempts to analyse data (Pb in honey) burdened with high load of noise.

Answer: We agree with the suggestion. In fact, we will conduct all our future research using ICP-MS. For the analysis of metals in honey, various methods of sample preparation can be applied (dilution, LLE, SPE, digestion, pyrolysis), followed by different instrumental techniques (AAS, ICP-OES, ICP-MS). However, when the elements under study exist in lower concentrations, high instrument sensitivity plays a huge role. Despite the proven effectiveness of ICP-MS in analyzing major and trace elements in honey, it represents significant investment that many laboratories might not be able to afford, as well as specialized environment with high purity gasses, which adds to the cost and the complexity of the technique. Energy-dispersive X-ray fluorescence (ED-XRF) as a reliable and non-destructive spectroscopy method has created significant interest over the last years, due to the availability of user friendly equipment at a reasonable price. Although, the detection limits are considerably higher than those achieved with classic analytical methods like AAS or ICP-MS, this ED-XRF drawback seems to have no impact on our study, since the obtained concentrations of target elements were well above these trace levels.

Minor remarks:

l111-112 can you please clarify what is the difference between “honey in combs (HC, mixed honey origin)” and other types of honey? What does mixed honey origin mean?

Clarification: Samples of honey in combs indicate that the honey has not been extracted; instead, it is harvested along with a portion of the honeycomb. This is typically done at the beginning of the season when the comb cells filled with honey are well-sealed, suggesting that a variety of nectar sources are available.

L112-114: add sentence: Samples of honey in comb are, in fact, honey that has not been extracted; instead, they are harvested along with a portion of the honeycomb.

l191 quality of Fig1. is low, and I would recommend to put all locations next to each other (in the same horizontal line)

Fig 1: The figure is attached again with higher quality. Locations were align horizontally.

l85 please add “lead” in front of Pb

L77: already lead put before Pb

l127 Pyrrolidine is missing one r ?

L129: corrected

Reviewer 2 Report

Comments and Suggestions for Authors

The authors present a study on mineral concentrations in different honey. It is an interesting study, giving a new perspective on honey analyses. However, major revisions are needed.

I would suggest to combine Results and Discussion sections if the journal allows it. As is now, results are repeated in discussion, or the are presented in the discussion and not in results. Ex. L257-258, L267-272…

The manuscript does not represent the title. The urban-rural gradient is not discussed.

Introduction:

L56-62: Citations are needed. Here are some suggestions:

Biol Trace Elem Res (2011) 140(2):170–176

Environmental Research (2022) 224, Part C: 112237

J Chem Ecol (2015) 41(4):386–395

r. PLoS ONE (2015) 10(7):e0132491.

Materials and Methods:

L107-111:  please specify the number of samples and when each honey type was taken. As it is written now it could be concluded that each honey type was collected at four different occasions during the season.

The usual method for measuring metals in digested samples is ICP-OES/MS. However, not all instrumental methods are available to all researchers.

L161-164: Standard statistical method for differences in elemental concentrations is usually ANOVA followed by Tukey’s post hoc test. Newman-Keuls post hoc test is usually used for social sciences.

Results:

L186-195: Both Table 1 and Figure 2 show the same data. I would suggest to use either one or the other. Graphical representation of the results is always nice and easier to follow. However, not all of the results can be represented in this way. Maybe use Figure and move the Table to supplementary.

L196-216: Tables 2-4 are not needed in the main text. I suggest to move them into supplementary. In addition, the results presented there are very confusing. Here you compare both the type of honey and location at the same time. Would it not make more sense to compare different honey types per location and see what elements show significant differences? And then repeat the same for each honey type and different locations. For example, take acacia honey and compare the differences between locations. Also, you could compare different location independent of the honey type. Obviously, there is a temporal aspect to the study, as samples were taken at four different time points. Maybe also do an analysis on this. However, then it should be mentioned that at each time point the honey type was different.

L225-229: Table 5 should also be moved to supplementary. The paragraph above nicely explains the results from the table.

Discussion:

The discussion section needs to be expanded. At the moment it is mostly focused on comparison with previous studies. The potential sources and reasons behind the findings are only discussed very shortly for Pb, while other elements are completely ignored in this regard.

Specific comments:

L267-277: Mean concentrations of Cr between comb honey (0.65 mg/kg) and meadow honey (1.46 mg/kg) at locations 1 were significantly different (Table 2.), with more than twice the difference between them. Why was this characterized as “relatively consistent”?

The same can be observed for Mn. Chestnut honey from location 3 had much lower concentration (0.80 mg/kg) compared to the same honey type from location 2 (2.01 mg/kg). This is represented in Table 3., but is not at all discussed in the discussion section.

L258-260: Change “certain” to “most”

L282-286: What honey samples from your study (location and type) are you comparing to studies done by Al Naggar et al and Demaku et al.? Please clarify and add further explanations.

L299-302: Here it is stated that Cu showed no statistically significant differences between locations and honey types. However, looking at Table 1. this statement seems very improbable. Chestnut honey at location 1 had a concentration of 3.01 mg/kg (with a very low SD of 0.08), and the same honey type had a concentration of 0.87 mg/kg (again with the very low SD of 0.09) and at location 3 of 0.98 mg/kg (SD of 0.02). It is very unlikely that location 1 does not have statistically higher concentrations, based on the reported values.

L313: One Zn should be removed.

L322-324: If a hive is relatively distant from a highway, then the possibility of traffic pollution should be lower not greater?

Conclusion

L344-346: There is no discussion about the urban-rural distribution of elements in the manuscript.

After the manuscript has been changed according to reviewers’ comments. The conclusions should also be changed.  

Author Response

The authors present a study on mineral concentrations in different honey. It is an interesting study, giving a new perspective on honey analyses. However, major revisions are needed.

I would suggest to combine Results and Discussion sections if the journal allows it. As is now, results are repeated in discussion, or the are presented in the discussion and not in results. Ex. L257-258, L267-272…

Answer: Results and Discussion are combined together. Discussion text from lines 245 to 340 is interspersed with the results. For example line 245-250 is inserted at the beginning of the introduction to line 179-185, and so on. The parts from the discussion are thrown in red. Added sentences are presented in purple. According to the changes in the discussion, the order of the reference numbers was also changed

The manuscript does not represent the title. The urban-rural gradient is not discussed.

Answer: we agree with the reviewer's comment and accordingly we will change the title of the paper, which is actually more precise regarding geographical locations: Mineral concentrations in different types of honey originating from three regions of continental Croatia

Line 86: sentence is modifeid with adition „assess potential variations in the levels“ and „and location near urban industrial zone“.

87-89: sentence is deleted.

Introduction:

L56-62: Citations are needed. Here are some suggestions:

Biol Trace Elem Res (2011) 140(2):170–176

Environmental Research (2022) 224, Part C: 112237

J Chem Ecol (2015) 41(4):386–395

  1. PLoS ONE (2015) 10(7):e0132491.

L56-72: we added references already been cited under: 5, 6, 13

Materials and Methods:

L107-111:  please specify the number of samples and when each honey type was taken. As it is written now it could be concluded that each honey type was collected at four different occasions during the season.

Answer: Three samples were taken from each apiary from three differently positioned hives, and samples were taken after the pasture of different melliferous plants. Due to that, we analyzed three different types of monofloral honey and honey from combs. During the season different plants are blooming, and according to nectar botanical resources, produced honey is a different botanical type. It means that those three different honey types were camplet et different seasone…So there is no temporal aspect within same toney types…

L95-96: To avoid misunderstanding sentences in modified to: A total of 36 honey samples, including meadow, acacia, chestnut, and mixed honey in combs (12 samples per location, 3 per type),

L108-109: “On each apiary samples were taken from three randomly chosen standard Langstroth hives (LR hives). During the active beekeeping season (May-July; September) four types of honey were sampled from the peripheral frames of the...“

The usual method for measuring metals in digested samples is ICP-OES/MS. However, not all instrumental methods are available to all researchers.

Answer: We agree with the suggestion. In fact, we will conduct all our future research using ICP-MS.

L161-164: Standard statistical method for differences in elemental concentrations is usually ANOVA followed by Tukey’s post hoc test. Newman-Keuls post hoc test is usually used for social sciences.

Answer: We agree. However, regarding the number of samples, the less conservative Newman-Keuls test was used.

Results:

L186-195: Both Table 1 and Figure 2 show the same data. I would suggest to use either one or the other. Graphical representation of the results is always nice and easier to follow. However, not all of the results can be represented in this way. Maybe use Figure and move the Table to supplementary.

Figure 2: we agree with the suggestion. In fact, considering that the results are listed in the table, we think that the picture is not really needed. In our opinion, it is more clearly visible from the tables. The picture is just a graphic processing... Therefore, we remove Figure 2  from the article.

L196-216: Tables 2-4 are not needed in the main text. I suggest to move them into supplementary. In addition, the results presented there are very confusing. Here you compare both the type of honey and location at the same time. Would it not make more sense to compare different honey types per location and see what elements show significant differences? And then repeat the same for each honey type and different locations. For example, take acacia honey and compare the differences between locations. Also, you could compare different location independent of the honey type. Obviously, there is a temporal aspect to the study, as samples were taken at four different time points. Maybe also do an analysis on this. However, then it should be mentioned that at each time point the honey type was different.

L203-220: Statistics showed in table 2 are explained in text and incorporated in Table 1. Therefore, tables 2-4 is excluded from article. The statistical results have been verified. The results for Cu, which were omitted by mistake, have been inserted. Statistics showed also comparison of different honey types per location. In Table 1 are added a- Significant differences between honeys within location; b- Significant differences in same type of honeys between locations. Statistical results were added in text: „Statistical analysis revealed significant differences in the mean concentrations of four honey samples, specifically for Cr, Cu, Mn and Fe. For Co, Pb, and Zn no significant differences were found. Honey from combs located in locations 2 and 3 exhibited significantly higher levels of Cr compared to honey from location 1. Additionally, a significantly higher concentration of Cr was found in meadow honey (1.46 mg/kg) compared to honey from the comb (0.65 mg/kg). Statistically significant differences in Mn concentrations were observed at location 1, where chestnut honey (1.13 mg/kg) had a significantly higher concentration than acacia honey (0.69 mg/kg). Conversely, at location 2, acacia honey exhibited higher concentrations (0.95 mg/kg) compared to chestnut honey (0.80 mg/kg). Furthermore, significantly higher concentrations of Mn were determined in chestnut honey at location 2 (2.01 mg/kg) compared to those from locations 1 and 3. Significantly elevated concentrations of Cu and Fe were found in chestnut honey at location 1 (3.01 mg/kg and 10.1 mg/kg, respectively) compared to locations 2 (0.87 mg/kg and 5.15 mg/kg) and 3 (0.98 mg/kg and 5.65 mg/kg). The concentrations of Cu and Fe in chestnut honey from location 1 were significantly higher than those in the other three types of honey from that location.“

Explanation: regarding the comment: Would it not make more sense to compare different honey types per location and see what elements show significant differences? We answer: During the season different plants are blooming, and according to nectar botanical resources, produced honey is a different botanical type. It means that those three different honey types were camplet et different seasone…So there is no temporal aspect within same toney types… To avoid misunderstanding sentences in modified to: “On each apiary samples were taken from three randomly chosen standard Langstroth hives (LR hives). During the active beekeeping season (May-July; September) four types of honey were sampled from the peripheral frames of the...“

L225-229: Table 5 should also be moved to supplementary. The paragraph above nicely explains the results from the table.

Answer: Regarding to explanations of multiple regression analysis results in text we decided to remove Table 5.

Discussion:

The discussion section needs to be expanded. At the moment it is mostly focused on comparison with previous studies. The potential sources and reasons behind the findings are only discussed very shortly for Pb, while other elements are completely ignored in this regard.

L304-313: Answer for explanation for obtained results sentences are added (senetnces is in purple letters): Previous studies have suggested a significant influence of soil composition on the microelement content in honeys [41]. However, the geochemical composition of soil in the studied locations in continental Croatia showed no significant differences in content of Co, Cr, Cu, Fe, Mn and Zn [42]. Statistical analysis indicated that only the levels of Cr and Zn in honey were significantly influenced by the location and type of honey. However, in this study, botanical origin has the most significant influence on element concentrations. The influence of geographical origin was only observed in the concentrations of Cu, Fe, and Mn in chestnut honey.

Specific comments:

L267-277: Mean concentrations of Cr between comb honey (0.65 mg/kg) and meadow honey (1.46 mg/kg) at locations 1 were significantly different (Table 2.), with more than twice the difference between them. Why was this characterized as “relatively consistent”?

The same can be observed for Mn. Chestnut honey from location 3 had much lower concentration (0.80 mg/kg) compared to the same honey type from location 2 (2.01 mg/kg). This is represented in Table 3., but is not at all discussed in the discussion section.

L203-220: Statistics showed comparison of different honey types per location. In Table 1 are added a- Significant differences between honeys within location; b- Significant differences in same type of honeys between locations. Statical results and interpretation is added were added in text:   

L258-260: Change “certain” to “most”

L250: “certain” is corrected to “most”

L282-286: What honey samples from your study (location and type) are you comparing to studies done by Al Naggar et al and Demaku et al.? Please clarify and add further explanations.

L272-276: we removed the above and inserted a comparison with works that are more similar to ours and replaced the references under 31 and 32 with new ones; Alda-Garcilope et al., 2012 and Oroian et al., 2016

L299-302: Here it is stated that Cu showed no statistically significant differences between locations and honey types. However, looking at Table 1. this statement seems very improbable. Chestnut honey at location 1 had a concentration of 3.01 mg/kg (with a very low SD of 0.08), and the same honey type had a concentration of 0.87 mg/kg (again with the very low SD of 0.09) and at location 3 of 0.98 mg/kg (SD of 0.02). It is very unlikely that location 1 does not have statistically higher concentrations, based on the reported values.

L214-220: evaluation is added: Significantly elevated concentrations of Cu and Fe were found in chestnut honey at location 1 (3.01 mg/kg and 10.1 mg/kg, respectively) compared to locations 2 (0.87 mg/kg and 5.15 mg/kg) and 3 (0.98 mg/kg and 5.65 mg/kg). The concentrations of Cu and Fe in chestnut honey from location 1 were significantly higher than those in the other three types of honey from that location.“

L313: One Zn should be removed.

L298: One Zn is removed.

L322-324: If a hive is relatively distant from a highway, then the possibility of traffic pollution should be lower not greater?

L318-320: Sentences is corrected to: These values were also observed in honey sourced from rural areas where hives are situated at a considerable distance from highways, indicating that there is no evidence of pollution from internal combustion engine vehicles.

Conclusion

L344-346: There is no discussion about the urban-rural distribution of elements in the manuscript.

L339-351: Explanation: the term was not the best chosen term to explain the samples that were analyzed, so it was removed from the title and text. It was not commented on as such, but the results obtained were commented on.

After the manuscript has been changed according to reviewers’ comments. The conclusions should also be changed. 

L338-351: conclusions are changed to: This study provides insights into the distribution of trace elements in honey collected from central continental Croatia. In most cases, the concentrations of elements in the various types of honey from the three locations were relatively evenly distributed. Iron exhibited the highest concentration, while Pb had the lowest levels. Maximal values were distributed between meadow honey (Cr, Co, and Zn) and chestnut honey (Mn, Fe, and Cu), while the highest Pb values was detected in honey in comb. Furthermore, significant differences in the mean concentrations of Cr, Cu, Mn and Fe between four honey types were observed. Conversely, no significant differences were found for Co, Pb, and Zn. Most of the significant differences between the elements are related to chestnut honey. The highest levels of Fe and Cu were found in chestnut honey, while Co and Zn were elevated in meadow honey, and Pb was most concentrated in honey in combs from Varaždin County (location 1). The elevated levels of trace elements in honey may have beeen influenced by anthropogenic activities, including environmental and agricultural sources of contamination, and also soil composition in microlocations.

Although apiaries near highways and urban areas are more likely to accumulate higher levels of Pb compared to those in rural areas, this study found relatively similar levels of Pb in samples from all three sites. However, all measured Pb concentrations exceeded the MRL establish by EU, raising concerns about the toxicological risks to honeybee colonies. Given the toxic nature of Pb, obtained concentrations highlighting the importance of continuos monitoring its levels.

Other changes:

Abstract L16-24: changes presented in red

Conflicts of Interest: regarding the company affiliation in the manuscript the potential conflict of interests is clarificated and added in paper:

The authors declare no conflicts of interest. The authors declare no conflicts of interest. Author Višnja Oreščanin was employed by Oreščanin Ltd. She participated in the analysis and visualization of the data obtained in the study, without any other roles within the company. The remaining authors declare that the research was conducted in the absence of any commercial or financial relationships that could be construed as a potential conflict of interest.

Reviewer 3 Report

Comments and Suggestions for Authors

Dear Authors, I find this topic interesting and relevant to the readers.

Here i provide some minor comments and suggestions:

1) In Line 19 you mention three types of honey but you don't specify their botanical origin. In my opinion the types of honeys should be added in the abstact. 

2) Line 41-42. Please explain further

3) Line 59. The sentence needs amendments. Honeybees usually forage within a 3km range. In special cases they may reach 8-10km. Please correct

4) Line 81. You mention 4 types of honey but in abstract you mention 3.

5)In line 256 you refer to Northern part of Croatia but in line 345 you mention central Croatia. Which one is correct?

General comment- Please provide further information about the identification of the botanical origin of the samples. Did you apply pollen analysis or sensor analysis? 

Comments on the Quality of English Language

The quality of English Language is ok

Author Response

Dear Authors, I find this topic interesting and relevant to the readers.

Here i provide some minor comments and suggestions:

1) In Line 19 you mention three types of honey but you don't specify their botanical origin. In my opinion the types of honeys should be added in the abstact. 

Line 16-17: in sentene is added types of honeys (meadow, acacia, chestnut and honey in combs) and three types corrected to four types.

2) Line 41-42. Please explain further

Line 41-43: sentences is added: The mineral composition of honey is correlated with the color of the honey and its antioxidant properties. Additionally, the concentrations of minerals are associated with the electrical conductivity of honey.

3) Line 59. The sentence needs amendments. Honeybees usually forage within a 3 km range. In special cases they may reach 8-10km. Please correct.

Lines 60-62: sentence is corrected: As honey bees typically forage within a range of 3 kilometers from their apiaries, and in certain cases may extend their foraging distance to 8-10 kilometers, they can collect ....

4) Line 81. You mention 4 types of honey but in abstract you mention 3.

Line 16-17 (abstract): in sentence is added types of honeys (meadow, acacia, chestnut and honey in combs) and three types corrected to four types.

5) In line 256 you refer to Northern part of Croatia but in line 345 you mention central Croatia. Which one is correct?

Line 339: in order to avoid misunderstandings, continental Croatia has been put everywhere

General comment- Please provide further information about the identification of the botanical origin of the samples. Did you apply pollen analysis or sensor analysis? 

Answer: No. samples of honey were taken during harvesting honey immediately after blooming some of the main plant pastures (meadow; acacia; chestnut; and honey in combs after ripening of mix honey)

Round 2

Reviewer 1 Report

Comments and Suggestions for Authors

The authors made significant changes according to given suggestions.